# Effect of Dietary Rumen-Protected L-Tryptophan Supplementation on Growth Performance, Blood Hematological and Biochemical Profiles, and Gene Expression in Korean Native Steers under Cold Environment

**DOI:** 10.3390/ani9121036

**Published:** 2019-11-27

**Authors:** Jae-Sung Lee, Wahyu Priatno, Jalil Ghassemi Nejad, Dong-Qiao Peng, Jin-Seung Park, Jun-Ok Moon, Hong-Gu Lee

**Affiliations:** 1Department of Animal Science and Technology, Sanghuh College of Life Sciences, Konkuk University, Seoul 05029, Korea; foodleeking@gmail.com (J.-S.L.); wpriatno.wp@gmail.com (W.P.); jalilgh@konkuk.ac.kr (J.G.N.); pdq15689x@foxmail.com (D.-Q.P.); 2Team of an Educational Program for Specialists in Global Animal Science, Brain Korea 21 Plus Project, Sanghuh College of Life Sciences, Konkuk University, Seoul 05029, Korea; 3Institute of Integrated Technology, CJ CheilJedang, Suwon 16471, Korea; jinseung.park@cj.net (J.-S.P.); junokee@cj.net (J.-O.M.)

**Keywords:** rumen-protected L-tryptophan, growth performance, metabolites, glucose, gene expression

## Abstract

**Simple Summary:**

In this study, the effect of dietary rumen-protected L-tryptophan (RPT) supplement on growth performance, blood hematological and biochemical profiles, and gene expression was investigated in beef steers during a cold environment. We revealed that supplementation of 0.1% RPT incorporated into diet was beneficial owing to enhanced growth performance by increasing the ADG and glucose level, decreasing the feed conversion ratio, and maintaining homeostasis in immune responses in beef steers in a cold environment.

**Abstract:**

We assessed the growth performance, physiological traits, and gene expressions in steers fed with dietary rumen-protected L-tryptophan (RPT) under a cold environment. Eight Korean native steers were assigned to two dietary groups, no RPT (Control) and RPT (0.1% RPT supplementation on a dry matter basis) for six weeks. Maximum and minimum temperatures throughout the experiment were 6.7 °C and −7.0 °C, respectively. Supplementation of 0.1% RPT to a total mixed ration did not increase body weight but had positive effects of elevating average daily gain (ADG) and reducing the feed conversion ratio (FCR) on days 27 and 48. The metabolic parameter showed a higher glucose level (on day 27) in the 0.1% RPT group compared to the control group. Real-time PCR analysis showed no significant differences in the expression of muscle (MYF6, MyoD, and Desmin) metabolism genes between the two groups, whereas the expression of fat (PPARγ, C/EBPα, and FABP4) metabolism genes was lower in the 0.1% RPT group than in the control group. Thus, we demonstrate that long-term (six weeks) dietary supplementation of 0.1% RPT was beneficial owing to enhanced growth performance by increasing the ADG and glucose level, decreasing FCR, and maintaining homeostasis in immune responses in beef steers in a cold environment.

## 1. Introduction

The climate in the Korean peninsula is becoming more polarized owing to the global climate change resulting in longer, colder winters and longer, hotter summers [1,2]. Given this inevitable phenomenon, the decline in productivity of ruminants resulting in serious economic effects due to low temperature in winter is expected to accelerate [3,4]. Therefore, it is pivotal to establish a nutritional strategy that helps to minimize the impaired productivity of beef cattle due to temperature conditions. Cold stress causes negative effects on growth performance and immune cell population by influencing metabolic and immunological activities [5,6]. Thus, hematological profiles, if measured, could be a good indicator of how stressors affect the immune function of the body [6,7,8,9]. The thermo-neutral zone for growing Korean native beef cattle is approximately 4 to 20 °C, and the critical temperature is approximately −10 °C [1,3]. In Korea, the temperature from Nov to Feb often falls below −20 °C at night and fluctuates from approximately −5 °C to −15 °C during the daytime. Given this, adaptation to cold environments in ruminants involves increasing thermal insulation, appetite, and basal metabolic intensity that improve cold hardiness and reduce the risks of both acute and chronic cold stress [5,6,10]. Cold stress affects beef cattle production by elevating resting heat production, the energy requirement for maintenance, and appetite drive, while decreasing feed digestibility [5,7]. The appetite stimulation may partially counteract the increased energy requirement, but it cannot fully alleviate the reduced efficiency of utilization of dietary energy [5,7,10].

L-tryptophan (TRP) is known to play a pivotal role in metabolic, physiological, and organ development and growth [11,12]. Lack of TRP may adversely affect feed intake and growth performance. However, in most ruminants, TRP is not deficient due to ruminal fermentation [11,13]. In particular, TRP in ruminants has been reported to increase the secretion and activity of pancreatic α-amylase, thereby increasing starch digestibility by promoting the secretion of the intestinal hormone cholecystokinin (CCK) [11,14,15]. TRP is also known as a constituent of niacin, a precursor of serotonin and melatonin, and to have antioxidative and stress-relieving properties [11,13]. Thus, TRP is proposed as an essential amino acid (AA) that can help avert the decline in animal productivity due to a low temperature environment under climate change [11]. On the other hand, the amounts of essential AAs, such as TRP, supplied by microbes are sufficient only to support maintenance and normal milk and beef production and not maximal animal growth [10,12]. Thus, in order to ensure the highest possible productivity in ruminants, it is essential to supply more AAs, in particular TRP, to meet the requirement.

Feeding dietary rumen-protected L-tryptophan (RPT) has been shown to not only increase growth performance of lambs [16] and wool output in sheep [17] but also improve N utilization including urinary N excretion and retained N, and average daily gain (ADG) in cashmere goats [13]. However, the effect of RPT on performance, blood hematology, and biochemistry and related gene expression in Korean native steer under cold stress conditions is yet to be investigated. Information regarding ruminal bypass AA has been limited to a few essential AAs but not RPT. Therefore, we hypothesized that dietary supplementation of RPT might enhance growth performance, alter blood parameters, and related gene expression in Korean native steers under cold environment, which was investigated in the present study.

## 2. Materials and Methods

### 2.1. Animals and Feeding Trial

The experimental procedure and methods were approved by the Animal Welfare and Ethics Authority of Konkuk University, Seoul, Republic of Korea (approval no: KU18178).

Eight Korean native steers [249 ± 21.6 day-old; body weight (BW) = 279 ± 16.6 kg] were selected and randomly assigned into one of two groups: total mixed ration (TMR) without RPT (control, n = 4) and TMR with 0.1% RPT (RPT, n = 4) groups. The experiment was conducted outdoors, under a shed, at the experiment farm (Chungju, Chungcheongbuk-do, Korea). All animals were protected from the rain by covering using a ceiling. The mean weights of the animals at the starting period of the experiment (on day 0) were not significantly different between the groups (*p* = 0.950). After grouping by statistical analysis of the body weights of cattle in each group, the animals were housed in 1.0 × 6 m^2^ individual pens for the duration of the experiment. TMR (Nonghyup Feed Co., Ltd., Yangju-si, Korea), which did not contain antibiotics, was fed to animals as a basal diet. The animals were fed 10 kg TMR/day/animal to meet the nutrition requirements of the National Research Council [18]. The RPT group was fed TMR mixed with RPT (0.1% of TMR as-fed basis; CJ CheilJedang, Suwon, Korea) once a day at 0800 h. The rate of bypass is above 95% (flowing corporation data). Water was available ad libitum. The chemical analysis (moisture, crude protein, crude fat, contents of 19 AAs, etc.) of the feed stored at −20 °C was performed (Table 1). Body weight of animals was measured three times, on day 0, 27, and 48. The ADG of the animals was calculated by dividing the difference in weight between the initial weight and the end weight by the total number of days during the experiment. Feed intake (FI) was calculated by examining total daily feeding and the orts (residual feed) amount of individual animals every morning at 0700 prior to providing fresh feed.

### 2.2. Environmental Qualifications and Measurements

Temperature (°C) and relative humidity (%) were recorded during the experiment period with a portable temperature and humidity meter (MHT-381SD, Lutron Co., LTD., Taipei, Taiwan) at hourly intervals (Table 2). As aforementioned in the introduction, the thermo-neutral zone for growing Korean native beef cattle is 4 to 20 °C. In the present study, total average temperature during the experiment period (days 7 to 48) was −0.3 °C.

### 2.3. Blood Collection and Analysis

Blood samples were collected three times on day 0, 27, and 48 of the experiment period at 1400 h from jugular venipuncture into K_2_ EDTA-treated vacutainers (4 mL; Becton Dickinson, Franklin Lakes, NJ, USA) for measuring hematology (white blood cells, lymphocytes, monocytes, granulocytes, red blood cells, hemoglobin, hematocrit, mean corpuscular hemoglobin, and plateletcrit) using the VetScan HM2 analyzer (Abaxis, Union City, CA, USA). Subsequently, blood in serum tubes was centrifuged at 2500× *g* for 15 min at 4 °C. The serum was used for measurement of biochemical parameters (glucose, total protein, blood urea nitrogen, albumin, and triglyceride) using the automated biochemical analyzer (Hitachi Automatic Analyzer model 7180, Hitachi, Tokyo, Japan) according to manufacturer’s instructions.

### 2.4. Tissue Collection and Analysis

Longissimus dorsi muscle sample (12 to 13 ribs) for each steer was collected by biopsy at the final day of the experiment (day 48) using a spring-loaded biopsy instrument (Biotech, Nitra, Slovakia). The tissue samples were washed in Diethylpyrocarbonate (D5758, Sigma Aldrich, St. Louis, MO, USA) and autoclaved water, transferred into a 2 mL tube, and then incubated in a liquid nitrogen gas locker until analysis. The tissue samples (2 g) were ground into powder under freezing conditions, with liquid nitrogen, and RNA was extracted from each tissue sample, as previously described [19]. The concentration of RNA was determined by spectrophotometric analysis (Nanodrop 1000, Thermo Scientific, Seoul, Korea). The RNA integrity was estimated using an RNA Nano 6000 Assay Kit for an Agilent Bioanalyzer 2100 system (Agilent Technologies, Inc., Richardson, TX, USA). When the number of RNA integrity (RIN) was more than 6, complementary DNA (cDNA) synthesis was performed. RIN in the current study was 6.5 ± 0.27. RNA was used for real-time PCR analysis.

To synthesize cDNA in the procedure of real-time PCR analysis, as previously described [19], 1 μg of RNA was reversely transcribed in a 100 μL reaction volume with an iScript ^TM^ cDNA synthesis kit (Bio-Rad Laboratories, Inc., Foster City, CA, USA) according to manufacturer’s instructions. Quantitative real-time PCR (qRT-PCR) was performed on duplicate samples by using a CFX Connect ^TM^ Real-Time System (Bio-Rad Laboratories, Inc.) with IQ ^TM^ SYBR Green Supermix (Bio-Rad Laboratories, Inc.) reagents. The following PCR conditions were used: 95 °C for 3 min and 40 cycles at 95 °C for 10 s, 51 °C to 65 °C for 30 s and 72 °C for 30 s. Following amplification, a melting-curve analysis was performed to verify the specificity of the reactions. The cycle threshold value was used as the PCR threshold cycle number for the end point used in the qRT-PCR quantification. A relative gene-expression analysis was implemented according to the gene study, by using Bio-Rad CFX Manager 3.1 software (Bio-Rad Laboratories, Inc.), with multiple housekeeping genes [19]. Expression levels of genes involved in muscle (MYF6, MyoD, and Desmin) and fat (PPARγ, C/EBPα, and FABP4) were determined. Triple housekeeping genes (18S, GAPDH, and RPLP0) were used as the internal controls. Primer sequences used for qRT-PCR assay are presented in Appendix A. The specific primer of genes involved in muscle (MYF6, MyoD, and Desmin) and fat (PPARγ, C/EBPα, and FABP4) is shown in Appendix A.

### 2.5. Statistical Analysis

Data sets were analyzed using the MIXED procedure of SAS (version 9.0; SAS Institute Inc., Cary, NC, USA). Whole blood for hematological analysis was collected three times (on day 0, 27, and 48). This data was used for repeated measurement analysis to investigate the interaction effect (treatment × day). Growth performance including BW, ADG, and feed conversion ratio (FCR), and mRNA expression were assessed using the Student’s *t*-test by using JMP 5.0 software package (SAS Institute Inc., Cary, NC, USA). The Tukey test was used to compare the differences between treatment means. The normality of the data distribution was tested prior to the final comparison by SAS. Statistical differences were considered significant at *p* < 0.05.

## 3. Results

### 3.1. Growth Performance During Cold Temperatures

We investigated performance parameters of ADG, FCR, and feed intake of steers supplemented with TMR containing 0.1% RPT during a cold environment. As shown in Table 3, higher ADG and lower FCR (both *p* < 0.05) in the RPT group compared with the control group were observed on day 27 of the experiment (Table 3). In addition, animals in the RPT group showed higher (*p* = 0.001) ADG and lower (*p* = 0.001) FCR compared with the control group during the final day of the experiment.

In the observation of feed intake (FI) in steers during cold temperatures, there was no difference in FI between the two groups during adaptation period (days 0 to 6); however, the FI was higher (*p* = 0.038) in the RPT group than in the control group during the final week (Table 4). Average FI in the RPT group was higher (*p* = 0.011) compared to that in the control group.

### 3.2. Physiological Parameters in Blood During Cold Temperatures

Blood hematological and biochemical parameters were measured to determine the physiological conditions of steers supplemented with TMR containing 0.1% RPT during the cold environment (Table 5).

Our results showed that 0.1% RPT did not cause blood hematological changes except for monocytes (MON; *p* = 0.005). Compared to the control group during the adaptation period (day 0), a higher value of MON in the control group was observed on day 48 (*p* < 0.05). In contrast, animals in the 0.1% RPT group did not show any changes in the value of blood MON during the whole experiment period. All values of MON in the control group and 0.1% RPT group were within the normal range.

All blood biochemistry parameters except glucose (*p* < 0.005) demonstrated no significant differences (*p* > 0.05) between the two groups. In this study, higher (*p* < 0.05) glucose level in the RPT group, compared with the control group, was observed on day 27 of the experiment period, and the glucose level returned to the normal values on day 48 of the experiment.

### 3.3. Relative Gene Expression in Fat and Muscle Loin Tissues of Steers During Cold Temperatures

We further observed changes in gene expression related to muscle and fat metabolism in *longissimus dorsi* tissues of steers supplemented with TMR containing 0.1% RPT during the cold environment (Table 6).

In the present study, there were no differences (*p* > 0.05) in the expression of muscle metabolism genes, including MYF6, MyoD, and Desmin, between the two groups. However, the expression of fat metabolism genes, including PPARγ, C/EBPα, and FABP4, was higher (*p* < 0.0001) in the RPT group compared to the control group.

## 4. Discussion

L-tryptophan (TRP) is reported to be a limiting AA in growing lambs [16] and cattle [11] during the process of non-protein N utilization. Thus, supplementation of TRP in a rumen-protected form has positive effects on growth performance. L-tryptophan metabolites can affect growth, development, and health of beef cattle [11]. It is also known as a precursor of a stress-relieving neurotransmitter called serotonin [11,20]. Therefore, in the present study, we can postulate that animals in the RPT group were more tolerant of the cold stress situation, and thus showing higher ADG and lower FCR.

We observed higher FI in the RPT group compared to the control group (Table 4). Ma et al. [13] supplied two dosages of rumen-protected TRP to cashmere goats and observed an increase in the final BW and ADG in the supplemented group. Higher total FI intake could also postulate the fact that the steers in the RPT group tended to consume more feed in order to receive higher amounts of RPT, enabling them to better cope with cold stress conditions during the experiment period. On the contrary, higher amounts of RPT (over 0.5% of TMR, dry matter (DM)) have been documented to cause decreased FI due to lower palatability [5,10,21]. However, no changes in FI was observed by supplementing 0.1% RPT in Korean native steers in a previous study in our laboratory [11] under normal environmental conditions. In contrast, in the present study, higher FI in the RPT group (Table 4) implies that the amount of RPT up to 0.1% did not decrease the palatability of the feed. As stated before, tryptophan is known as a precursor of a stress-relieving neurotransmitter called serotonin [11,20]. Given this phenomenon, tryptophan supplementation may indirectly and positively help the animals to deal with cold stress in this experiment and thus indirectly induce FI in the tryptophan supplemented group.

In this study, higher blood glucose of cattle in the RPT group, compared with the control group, was shown on day 27 of the experiment period, and the glucose level returned to the normal values on day 48 of the experiment (Table 5). Abeni et al. [22] stated that decreasing glucose content in blood in heat-stressed lactating Friesian cows could be the result of reduced energy intake. They also suggested that heat stress negatively affects gluconeogenesis as an endocrine acclimation to the stress conditions [22]. L-tryptophan has been suggested not only to upregulate gluconeogenesis in the liver but its metabolites, including serotonin, modulate glucose uptake into muscle and thus suppress the rise in glucose. However, the reason why blood glucose was higher in the RPT group, to the authors, is unknown. In this study, we postulated that higher glucose levels in blood provides sufficient energy for thermoregulatory purposes in order to better dealing with cold stress.

In a previous study in our laboratory [11], we observed that relative mRNA expression levels, including MYF6, MyoG, FABP4, and LPL genes, were higher in the RPT supplemented group than those in the control group in Korean native steers. While the MYF6 and MyoG are representative of muscle differentiation [23], FABP4 is documented to be involved in intracellular transport and fatty acid metabolism [24,25]. Thus, 0.1% RPT alone may not be effective in altering the expression of muscle metabolism genes, including MYF6, MyoD, and Desmin. However, with respect to the fat metabolism and its related genes, 0.1% RPT could decrease the expression of PPARγ, C/EBPα, and FABP4 genes. The putative role of C/EBPα on nitrogen, glucose, and lipid metabolism is well documented [26,27]. C/EBPα is one of the main transcriptional mediators of the early stage of adipogenesis [28,29]. A study on leptin-deficient mice suggested that hepatic C/EBPα positively regulates lipogenesis in the mice [27]. They suggested that C/EBPα may regulate nuclear factor Y (YF-Y) or Sp1, whereas the NF-Y consensus sequence CCAAT is included in most genes involved in lipogenesis, including farnesyl diphosphate synthetase (FPP) and 7-dehydrocholesterol reductase (7DCR). Loss of expression of C/EBPα resulted in attenuated induction of typical lipogenic genes and FPP and 7DCR, in the leptin-deficient mice [27]. On the other hand, acetyl-CoA and NADPH required for fatty acid synthesis and β-oxidation were dysregulated in the liver of Δ proline-histidine rich domain (Δ PHR), but not in adipose tissue, which resulted in increased hepatic triglyceride production [26]. They found that distinct C/EBPα motifs regulate lipogenic and gluconeogenic gene expression in mice. It is thought that FABPs roles include fatty acid uptake, transport, and metabolism. The decrease in gene expression of PPARγ, C/EBPα, and FABP4 in the present study may be attributed to the muscle tissues that utilized free fatty acid for myocyte differentiation. This phenomenon could be caused by direct involvement of TRP as a blocking block or by indirect involvement of metabolic components related to TRP or both. Given the above review, it can be suggested that dietary supplementation of RPT may alter intracellular transportation of fatty acids by inhibiting the catabolism of fat in muscle. The intramuscular fat and intramuscular fatty acid concentration are important in meat quality improvement [30,31]. It has been stated that peroxisome proliferator-activated receptor γ (PPARγ) is the pivotally important gene in relation to lipid metabolism in muscle tissue [32]. Recently, Yang et al. [33] investigated the effects of diets with different energy levels on fat deposition and the fatty acid profile of the *longissimus dorsi* muscle in yak. They concluded that the high energy diets promoted the deposition and partial fatty acid content of *longissimus dorsi* muscle mainly by up-regulation of mRNA expression of ACACA, SCD, FASN, SREBP-1c, PPARγ, and FABP4. However, in the present study, since the energy and amino acid are adversely related, up-regulation of mRNA expression of PPARγ, C/EBPα, and FABP4 could be seen in the non-supplemented RPT group (control), which is in line with the aforementioned study. Different energy or protein levels, herein the supplementation of TRP, may alter intramuscular fat deposition into muscle by regulating PPARγ. PPARγ is in charge of some promotion, including adipocyte proteins or enzymes such as fatty acid binding protein (FABP4), fatty acid synthase (FASN), and lipoprotein lipase (LPL) [34]. Since very little information is available regarding the effect of RPT on fatty acid gene expressions in *longissimus dorsi* muscle, further investigations are necessary in order to confirm these results and to bring more insights to the available knowledge.

## 5. Conclusions

This study indicates that long-term (six weeks) dietary supplementation of 0.1% RPT enhances growth performance in Korean native steers by modulating the immune responses and elevating glucose levels under cold environment. In addition, 0.1% RPT reduced adipogenic gene expression, which may contribute to muscle tissues that utilized free fatty acid for myocyte differentiation of steers during the cold environment. Therefore, dietary supplementation of 0.1% RPT is beneficial in reducing the decline in productivity of beef cattle during cold stress.

## Figures and Tables

**Table 1 animals-09-01036-t001:** Chemical composition and amino acids of the basal diet used in this study.

	TMR ^1^
Chemical composition, % on a dry matter basis	
Dry matter	86.82
Crude protein	15.05
Ether extract	2.49
Crude fiber	25.90
Crude ash	6.75
Acid detergent fiber	32.87
Neutral detergent fiber	52.26
Amino acids, % on a dry matter basis	
Tryptophan	0.09
Methionine	0.09
Niacin	0.00
Lysine	0.54
Aspartic acid	1.20
Threonine	0.48
Serine	0.57
Glutamic acid	2.04
Glycine	0.61
Alanine	0.79
Valine	0.67
Isoleucine	0.47
Leucine	0.97
Tyrosine	0.29
Phenylalanine	0.55
Histidine	0.29
Arginine	0.71
Cystine	0.12
Proline	0.91

^1^ TMR, total mixed ration.

**Table 2 animals-09-01036-t002:** Changes in the temperature and relative humidity during the experiment period.

Period, Days	Temperature, °C			Relative Humidity, %		
	Min	Max	AVG ^1^	Min	Max	AVG
0 to 6 ^2^	−2.4	9.7	1.3	19.4	94.2	69.4
7 to 13	−15.7	13.4	−7.0	12.2	78.1	45.3
14 to 20	−14.0	13.5	−5.4	10.3	85.9	45.6
21 to 27	−14.6	15.8	−3.2	12.1	87.1	51.3
28 to 34	−7.3	29.8	1.9	17.9	85.3	45.9
35 to 41	−6.9	22.0	3.7	17.9	89.8	50.8
42 to 48	−5.9	23.5	6.7	15.8	89.9	61.9
7 to 48	−9.5	16.8	−0.3	15.1	87.2	52.9

^1^ AVG, average of temperature and relative humidity during day and nighttime for 7 days; ^2^ days 0 to 6, adaptation period.

**Table 3 animals-09-01036-t003:** Effect of 0.1% L-tryptophan (RPT) supplementation added to total mixed ration (TMR) on growth performance in cattle during a cold environment.

	Control ^1^	RPT ^2^	SEM ^3^	*p*-Value
Day 0				
Body weight, kg	278.5	279.7	7.69	0.950
Day 27				
Body weight, kg	285.1	301.5	8.30	0.360
ADG ^4^, kg/day	0.224	0.753	0.130	0.030
Feed conversion ratio ^5^	44.4	13.3	13.25	0.035
Day 48				
Body weight, kg	290.9	315.7	4.72	0.194
ADG, kg/day	0.262	0.766	0.1	<0.001
Feed conversion ratio	13.03	4.82	1.18	<0.001

Values are expressed as means (n = 4). ^1^ Control, no RPT supplementation to TMR; ^2^ RPT, 0.1% RPT supplementation to TMR; ^3^ SEM, standard error of the mean; ^4^ ADG, body weight gain against initial body weight/experimental days; ^5^ Feed conversion ratio, ratio of total feed intake versus total weight gain.

**Table 4 animals-09-01036-t004:** Effect of 0.1% L-tryptophan (RPT) supplementation added to total mixed ration (TMR) on feed intake in cattle during a cold environment.

Period, days	Control ^1^	RPT ^2^	SEM ^3^	*p*-Value
0 to 6 ^4^	8.93	8.56	0.108	0.088
7 to 13	9.24	9.21	1.647	0.948
14 to 20	9.41	9.56	1.171	0.623
21 to 27	9.60	9.68	1.200	0.791
28 to 34	9.68	10.22	1.557	0.201
35 to 41	9.15	10.02	0.743	0.063
42 to 48	9.54	10.56	0.725	0.038
7 to 48	9.43	9.86	1.533	0.011

Values are expressed as means (n = 4). ^1^ Control, no RPT supplementation to total mixed ration; ^2^ RPT, 0.1% RPT supplementation to TMR; ^3^ SEM, standard error of the mean; ^4^ Days 0 to 6, adaptation period.

**Table 5 animals-09-01036-t005:** Hematological and biochemical analyses of blood in steers supplemented with TMR containing 0.1% RPT during a cold environment.

Items ^4^	Control ^1^			RPT ^2^			SEM ^3^	*p*-Value		
	D 0	D 27	D 48	D 0	D 27	D 48		Treatment (T)	Days (D)	T × D
Hematological parameters										
WBC	9.28	9.44	8.92	8.66	8.70	9.34	0.335	0.6815	0.9841	0.7830
LYM	6.07	6.25	5.26	5.55	6.55	5.98	0.234	0.7329	0.5086	0.3023
MON	0.07 ^b^	0.08 ^b^	0.56 ^a^	0.06	0.07	0.12	0.043	0.0047	0.0002	0.0019
GRA	2.98	2.94	3.44	3.01	2.09	3.29	0.214	0.4665	0.3074	0.4322
RBC	10.33	9.97	9.00	10.70	10.34	10.08	0.266	0.2118	0.2050	0.6769
HGB	13.33	13.73	12.65	13.60	14.20	13.68	0.237	0.2449	0.9520	0.4339
HCT	34.57	34.12	31.66	35.00	34.64	33.88	0.584	0.3946	0.3787	0.7956
MCH	13.03	13.85	14.10	12.83	13.80	13.70	0.248	0.6806	0.2577	0.9625
PLT	398.5	455.8	367.5	378.0	476.8	379.5	15.39	0.8862	0.0360	0.8263
Biochemical parameters										
Glucose	84.0	87.0	78.5	77.0 ^B^	86.3 ^A^	74.3 ^B^	1.32	0.0534	0.0015	0.4345
TP	6.28	6.28	6.05	6.40	6.48	6.20	0.055	0.1524	0.1444	0.9577
BUN	15.68	19.65	15.35	15.60	18.95	16.05	0.527	0.9773	0.0036	0.8052
Albumin	3.65	3.58	3.53	3.50	3.60	3.45	0.033	0.3378	0.4384	0.5805
TG	13.3	13.3	13.0	11.8	18.8	11.3	1.09	0.7354	0.3034	0.3293

Values are expressed as means (n = 4). ^a,b, A,B^ Indicate significant differences compared to the initial period in the control and RPT groups (*p* < 0.05, Tukey test); ^1^ Control, no RPT TMR; ^2^ RPT, 0.1% RPT supplementation to TMR; ^3^ SEM, standard error of the mean; ^4^ Abbreviations: WBC, white blood cell; LYM, lymphocyte; MON, monocyte; GRA, granulocyte; RBC, red blood cell; HGB, hemoglobin; HCT, hematocrit; MCH, mean corpuscular hemoglobin; PLT, platelet; TP, total protein; BUN, blood urea nitrogen; TG, triglyceride.

**Table 6 animals-09-01036-t006:** Relative expression of fat (PPARγ, C/EBPα, and FABP4) and muscle (MYF6, MyoD, and Desmin) genes in loin tissue of cattle supplemented with total mixed ration (TMR) containing 0.1% L-tryptophan (RPT) during the cold environment.

	Control ^1^	RPT ^2^	SEM ^3^	*p*-Value
PPARγ	1.000	0.274	0.2302	<0.000
C/EBPα	1.000	0.293	0.2714	<0.000
FABP4	1.000	0.233	0.2618	<0.000
MYF6	1.000	0.834	0.1667	0.071
MyoD	1.000	0.857	0.1822	0.676
Desmin	1.000	0.871	0.1902	0.140

Values are expressed as means (n = 4). Loin tissue of each cattle was used to estimate the expression of the fat (PPARγ, C/EBPα, and FABP4) and muscle (MYF6, MyoD, and Desmin) genes by real-time PCR analysis. The results were normalized using GAPDH, RPLP0, and 18S as internal controls; ^1^ Control, no RPT TMR; ^2^ RPT, 0.1% RPT supplementation to TMR; ^3^ SEM, standard error of the mean.

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
