# Peer review of "Effect of Dietary Rumen-Protected L-Tryptophan Supplementation on Growth Performance, Blood Hematological and Biochemical Profiles, and Gene Expression in Korean Native Steers under Cold Environment"

_animals, 2019, doi:10.3390/ani9121036_

Round 1

Reviewer 1 Report

All good.

Author Response

Dear sir. Reviewer,

Thank you very much for your positive consideration.

Sincerely,

Hong-Gu LEE, the corresponding author ([email protected])

Reviewer 2 Report

While most of my queries have been answered adequately, I still have concerns about the low sample size. Without a Power Analysis, I am not convinced that the sample size is adequate for the paper to draw its conclusions. If the authors can provide me with several examples of similar peer-reviewed studies that have used a similar sample size, then I will be happy enough to accept it in its current form.

Author Response

Dear sir. Reviewer,

Thank you very much for all valuable comments and positive consideration for publication.

Sincerely,

Hong-Gu LEE, the corresponding author ([email protected])

Reviewer 3 Report

Recently is more and more information about tryptophan which plays a role in energy expenditure. This type of studies are needed. In small animals model is proved that it is via GRP receptor. More studies in ruminant are needed.

Manuscript is clearly written and understood. The results are well presented.The discussion is well conducted.

Author Response

Dear sir. Reviewer,

Thank you very much for all valuable comments and your positive consideration.

Sincerely,

Hong-Gu LEE, the corresponding author ([email protected])

This manuscript is a resubmission of an earlier submission. The following is a list of the peer review reports and author responses from that submission.

Round 1

Reviewer 1 Report

The authors illustrate the physiological, biochemical and gene expression changes in tryptophan supplemented cattle under cold stress conditions. The present study design does not sufficiently support the hypothesis on growth performance, haematological and biochemical changes after tryptophan supplementation owing to lower sample size and observations. However, the gene expression changes are informative. 

Introduction

56-59-give reference

68-add reference

Materials and methods

The number of animals used is very low for assessment of ADG, feed conversion ratio, physiological and biochemical analyses. However, for gene expression profiling four samples is enough.

The whole experiment is conducted for 40 days and the sample collection and analysis is done only for 3 days (0, 27 and 40). Hence in the treatment and control groups only a total 12 measurements are taken. In order to increase the power of the experiment either increase the number of animals in each group and /or increase the number of observations from each animals.

Explain the protocol used for real time PCR and calculation of relative gene expression in detail.

Explain why muscle tissues were particularly selected for gene expression studies.

Explain the suitability of house keeping genes used in the study with references.

The effect of cold stress is not measured in both the control and in the treatment groups. There are biomarkers for stress tolerance. No assessment is done for stress biomarkers. Hence, cold stress and tryptophan supplementation cannot be correlated based on the present study.

Results

198- Delete the sentence. No significant changes in blood biochemical parameters in no way indicate stability of the homeostatic mechanisms.

216-change compare to compared

Discussion

223- It is not possible to conclude that RPT animals had better cold stress tolerance because no tolerance assay has been performed in the present study.

225-233- Since no measurement of hormones is done in the present study, the results of the present study do not justify the assumption that high feed intake could be due to tryptophan supplementation.

234-240- Present results are not sufficient to substantiate this.

241-255- Present results are not sufficient to substantiate this.

256-270-The role of glucose is extrapolated to explain the digestibility and metabolism in ruminants. The conclusions thus made based on this are not supported in the study.

271-309- The gene expression changes are informative with regard to tryptophan supplementation and its beneficial effects in steers.

Author Response

Dear sir. Reviewer,

Please find enclosed the revised version of our manuscript entitled “Effect of dietary rumen-protected L-tryptophan supplementation on growth performance, blood hematological and biochemical profiles, and gene expression in Korean native steers under cold environment”

We trust that we have made all of the changes necessary again and that the manuscript is now ready for publication.

Thank you very much for all valuable comments.

Sincerely,

Hong-Gu LEE, the corresponding author ([email protected])

Reviewer 2 Report

Comments to the Authors of manuscript number: animals-616584 entitled “Effect of Dietary Rumen-Protected L-Tryptophan Supplementation on Growth Performance, Blood Hematological and Biochemical Profiles, and Gene Expression in Korean Native Steers under Cold Environment”.

In the present paper I reviewed effects of the supplementation with rumen protected L-tryptophan on growth performance, blood hematology and biochemical analysis of the blood as well as gene expression in steers in low temperature are presented.

The study is very interesting. It requires a lot of efforts. Maintaining such large animals is very expensive as well as the analyzes performed.

 The number of animals in each group may optionally be accepted, although for good statistical analysis n=6 is considered.

Author Response

(The authors gave the same response as above.)

Reviewer 3 Report

Line 51-52: Could you further define "thermo-neutral" and "critical temperature".

Line 62: Please check this statement. Is TRP deficient or "not" deficient? The rest of the paragraph suggests it is not enough to counter cold stress and needs supplementation.

Line 74: What is N utilization? Please define further.

Line 86: Can you explain how you decided on the total sample number of 8 (4+4) for your experiment? Was any statistical test used?

Line 92: What does "grouping by statistical analysis of individual body weights" mean? Could you elaborate on this?

Line 102: "orts" seems to be a spelling error- what is the word meant to be?

Line 111: What is the "adaptation period" mentioned here, and what role did it play? Was the RTP given during this period or was it started on Day 7? Please explain this in further detail.

Line 133: Where was this previously described?

Line 165: Consider adding "added to TMR" in the Table heading to make its meaning clearer.

Line 174: P=0.005 but the table says p=0.011 for the average FI.

Line 187: Spelling error "Turkey" instead of "Tukey"

Line 200: What the does the word "expectedly" mean here? Why is it expected that Blood Glu would rise at Day 27 and then return to normal by the end of the experiment?

Line 204-205: Earlier it says that muscle biopsies were taken from the Latissimus Dorsi, however here, it mentions loin muscle. It is my understanding the these are different anatomical regions. Which is the correct region?

Line 218: While this sentence may be clear enough to readers better acquainted with ruminant nutrition, I would like a bit more explanation so I understand what this means in lay terms.

Line 223: Are you able to explain the difference between ADG and weight gain? The results seem to show there is no significant total weight gain between the groups but the difference in ADG is significant. In this sentence, however you mention the term "weight gain" as a significant finding, and I'm unsure of the exact difference between the two.

Line 225: Comma missing.

Lines 225-240: I suggest you revise these lines somewhat. There is no mention of what the link between serotonin levels and FI is, so the initial lines do not flow well. Both paragraphs also seem to have a repeat of the final postulation.

Lines 256-270: Similarly in these paragraphs, glucose levels are mentioned as stress indicators as well  as in relation to digestion modulation. It is not clear in what capacity the increased glucose levels are interpreted as relieving stress. I suggest a re-write of this section to make the clearer to the reader, exactly how the higher glucose levels are interpreted as adding a positive effect to the cattle.

Lines 271-309: Similarly in these paragraphs, while the technical information is important, I suggest that the gist of it be broken down into simpler language, to make your point easier to understand, perhaps a few extra lines at the end, explaining what the gene expressions mean for homeostasis,  and muscle/fat breakdown.

Author Response

(The authors gave the same response as above.)

Reviewer 4 Report

109 to 111. First, two time periods, minimum and maximum, that should be 4 values, not two, and neither of the numbers written here reflects the data in Table 2, the numbers you written here looks more like the average temperature with a forgotten negative sign in front of the 7. Next sentence, 9.5oC? This is again contradictory to your Table 2., where the average temperature in this period is -0.3oC. Science should strive for precise, concise writing, the information written between line 109 to 111 is not only incorrect, but it is redundant to information in Table 2, in this paragraph, it is best to just delete every word after (Table 2).
123. Biochemical parameters
130. delete "treated"
134. Nanodrop, not nanodrip
146. please justify the selection of house keeping genes, either by software such as Bestkeeper or by citing published literature that used GAPDH and RPLP0 as internal control
229, 296. please italicize the et al
280,281. C/EBPα gene encodes transcription factor called CCAAT..., and rather than explaining what is a trancription factor in the next sentence, instead, please elaborate on the role of CCAAT enhancer-binding protein alpha, what genes it regulates and how it affects lipogenesis, citing papers such as Pederson et al., 2007.
290. No. Your data does not suggest such conclusion, since your qPCR also showed a reduced relative expression of muscle differentiation marker genes in the RPT group, albeit not significant, so you must not say it enhanced muscle differentiation. I would suggest a rewrite on this part of the discussion entirely, focus on the interpretation of reduced relative expression of fat marker genes.

Author Response

(The authors gave the same response as above.)
